# A Combined Electromagnetic Induction and Radar-Based Test for Quality Control of Steel Fibre Reinforced Concrete

**DOI:** 10.3390/ma12213507

**Published:** 2019-10-25

**Authors:** Janusz Kobaka, Jacek Katzer, Tomasz Ponikiewski

**Affiliations:** 1Faculty of Civil Engineering, Environmental and Geodetic Sciences, Koszalin University of Technology, 75-453 Koszalin, Poland; janusz.kobaka@tu.koszalin.pl; 2Faculty of Geodesy, Geospatial and Civil Engineering, University of Warmia and Mazury in Olsztyn, 10-720 Olsztyn, Poland; jacek.katzer@uwm.edu.pl; 3Faculty of Civil Engineering Silesian University of Technology, 44-100 Gliwice, Poland

**Keywords:** SFRC, non-destructive testing, quality control, electromagnetic induction, radar, fibre

## Abstract

In this paper, the authors made an attempt to detect the fibre content and fibre spacing in a steel fibre reinforced concrete (SFRC) industrial floor. Two non-destructive testing (NDT) methods, an electromagnetic induction technique and a radar-based technique, were applied. The first method allowed us to detect the spacing in subsequent layers located in the thickness of the slab. The result of the second method was a 3D visualization of the detected fibre in the volume of concrete slab. The conducted tests showed aptitude and limitations of the applied methods in estimating fibre volume and spacing. The two techniques also allowed us to locate the areas with relatively low fibre concentration, which are very likely to be characterized by low mechanical properties.

## 1. Introduction

Despite its brittleness and low capacity to bear tensile stress [1,2], ordinary concrete is the most frequently used building material in the world. Concrete is almost always reinforced by some kind of steel elements (e.g., bars, stirrups, meshes) when used for construction to counteract its brittleness. Over the last 50 years, steel fibre has become a more and more popular type reinforcement for concrete. Currently, steel fibre reinforced concrete (SFRC) is one of the main building materials [3,4]. SFRC is characterized by higher tensile, shear and flexural strength [5,6,7,8], better performance when exposed to elevated temperatures [9] and lower shrinkage [10] than ordinary concrete of similar composition. Usually, steel fibre is arranged randomly but evenly in the SFRC volume. Nevertheless, in some circumstances, irregularity in the fibre distribution may occur. These irregularities in fibre spacing significantly affect its properties. The homogeneous spacing of randomly oriented fibre within a structural element is crucial for guaranteeing proper structural performance with a satisfactory degree of repeatability [11]. Many attempts are being made to study fibre spacing using different destructive and non-destructive methods. The most successful methods are X-ray tomography [12,13] and cross-section analysis [14]. Both methods give very precise results but are not feasible in common in-situ test scenarios. The simplicity of a conducted test and affordability of used equipment are two main factors influencing the practicability and popularity of a testing method. In many cases, construction companies already have apparatuses dedicated for non-destructive testing (NDT) location of steel bars in existing concrete structures. Harnessing such apparatuses for the assessment of steel fibre volume and spacing would be the most practical and sustainable solution. In the authors’ opinion, two existing NDT techniques, electromagnetic induction and radar-based techniques, have the potential for testing SFRC. The inductive technique was proven as a robust and simple non-destructive method to assess the content and the distribution of steel fibre. Nevertheless, there is still a necessity to define its accuracy. The equations for the conversion of the inductance into fibre volume and spacing [2] are also needed. Multiple attempts have also been made to use radar-based technique for the similar assesment of SFRC. The fundamental differences between electromagnetic induction and radar technologies are the detectability of objects of different materials and the dependency on the properties of the base material. Using the induction technology, only ferrous materials can be detected. Radar techniques allow the detection of objects of different materials, including ferrous and non-ferrous metals, water-filled pipes, voids, etc. [15]. Moreover, the radar-based technique can be applied not only to detect location of reinforcement, but also to assess moisture in reinforced concrete [16].

The authors decided to conduct a research programme focused on using both techniques simultaniously to assess fibre spacing in hardened SFRC. Combining two separate NDT methods proved to be very efficient in assessment of mechanical properties of concretes with no fibre reinforcement [17]. The results achieved this way were much closer to real strength characteristics than the results based on only one method. In the authors’ opinion, it is feasible to harness devices which are commercially available (and commonly used to detect rebars) to instantly detect fibre and assess their volume and spacing. The main aim of the conducted research programme was to prove the concept and enable further research in this area.

## 2. Materials and Equipment

The tests were carried out on the industrial floor located inside a depot building in Koszalin, Poland (see Figure 1). The tested floor (493 m^2^) was composed of two layers: A 100 mm-thick concrete undercoat and a 150 mm-thick main SFRC layer. Hooked steel fibre (dimensions of the fibre 50 × 1.0 mm, tensile strength 1115 MPa according to EN 14889-1 standard) at a volume of *V_f_* = 0.3% constituted the reinforcement. The declared strength class of concrete was C20/25. The tests were carried out six years after the concrete was casted.

Both NDT devices used in the research programme were developed for the assessment of rebar location in hardened concrete [18]. The measuring device using electromagnetic induction technique [19] was designed to scan a flat square area of 0.36 m^2^ [20]. The principle of the device’s operation is as follows: When alternating current runs through the probe coil of the device, an electromagnetic field appears around the coil. If there is a ferromagnetic material in the field, it brings about a change in the voltage of the coil, and the voltage change appears according to the diameter around the cover thickness of the rebar [21]. The method dedicated to the rebar detection was adopted in the tests to localize steel fibre in the area. Using the electromagnetic induction device, 32 randomly chosen square areas of the floor were tested. The total tested area was equal to 11.5 m^2^, which represented 2.3% of the area of the whole floor. The measuring device allowed to conduct tests in depth up to 150 mm. During the tests, four measuring depths were scanned: 30, 60, 90 and 120 mm.

The applied radar apparatus emitted radar pulses spread over the frequency range from 1.0 to 4.3 GHz [22]. The lower the frequency, the deeper the subsurface is penetrated, while the higher the frequency, the smaller objects can be spotted [15].

When the radar device is moved over the surface, a measurement is taken every 5 mm. At one scanner position, a high number of pulses are emitted and recorded to determine the full reflection pattern of the objects under the surface. Multiple acquisitions are used to reduce the noise in the data, which leads to a clean image [15].

The signal acquired by the radar front-end is further conditioned by the following steps:Correction of antenna sizes and positions;Background removal with automatic foreground/background detection to mask uniform structures such as the surface and possible stratifications;Automatic gaining to compensate the damping of the radar waves in the base material;Time-zero estimation (automatic recognition of the surface position);Temperature compensation to allow immediate and accurate measurements directly after start-up.

## 3. Test Results and Analysis

The exemplary images of the industrial floor tested using an electromagnetic induction technique are shown in the Figure 2. The obvious disadvantage of this scanning method is the “shadow” cast by the fibre present in the top layers. The created “shadow” increased blackening of the images in deeper layers. Therefore, in subsequent layers, the shaded areas should not be taken into account to assess fibre presence.

The advantage of the electromagnetic induction method is a possibility of instant assessment of fibre volume and spacing in the scanned area. Images of tested areas located at the depth of 30 mm, presented in Figure 3, are examples of such possibility.

The relative percentage of detected fibre in the layers located every 30 mm is presented in Figure 4. In subsequent layers, the shaded areas were not taken into account to assess relative percentage of fibre content. The electromagnetic induction technique detected almost 50% of steel fibre in the second layer located at the depth of 30–60 mm. The other layers contained from 15.5% to 20.2% of steel fibre. The low fibre content in the top layer can be explained by the so-called “wall effect”, which has been thoroughly described in literature [23,24,25]. The results obtained for the 0–30mm and 30–60mm depths closely corresponded to the actual fibre content. The smaller amount of fibre detected at depth of 60–120mm did not harmonize with genuine fibre content. The analysis of specimens was obtained by coring confirmed high uniformity of fibre spacing across the thickness of the floor (apart from the top layer). The explanation for this phenomenon might be a low measure of sensitivity of the apparatus. The depth and “shadow” cast by the fibre present in the top layers play a key role in this phenomenon.

The percentage deviation of fibre volume for 32 tested areas is presented in Figure 5. The results were grouped by depth. The top layer was characterized by the highest percentage deviation of fibre volume. The proximity of the surface influenced the homogeneity of fibre distribution. The second layer, placed at the depth of 30–60 mm, was characterized by the lowest value of the percentage deviation of fibre volume. Along with the depth, the value increased. The second layer, due to the best reading parameters and lowest percentage deviation of fibre volume, was the most suitable for assessing volume and spacing of steel fibre.

The results of fibre detection process of a square section (600 mm × 600 mm) of the industrial floor, based on the radar technique, are presented in Figure 6. The red colour indicates areas characterized by high concentration of fibre. The testing equipment was able to detect a high concentration of fibre only in the top layer, which can be observed in the bottom and right view of the image. The advantage of the method is the 3D image of the tested area, which can be rotated at any angle in the program supplied with the device (Figure 7).

## 4. Fibre Spacing Determination and Discussion

The three-parameter generalised beta distribution was chosen to define fibre spacing. This is the three-parameter distribution with the PDF (probability density function) described by the following formula:(1)$$g\left(x;p,\;\gamma ,\delta \right)=\frac{p}{B\left(\gamma ,\delta \right)}x^{\gamma p-1}{\left(1-x^{p}\right)}^{\delta -1}$$
in the formula, B(γ, δ) is beta function defined by the following equation:(2)$$B\left(\gamma ,\;\delta \right)=\frac{\Gamma \left(\gamma \right)\Gamma \left(\delta \right)}{\Gamma \left(\gamma +\delta \right)}$$
where Γ (gamma function):for positive integer numbers *n*:
(3)Γ(n)=(n−1)!
for other positive numbers *z*:
(4)$$\Gamma \left(z\right)=\int_{0}^{\infty }x^{z-1}e^{-x}dx$$

The main advantage of using the three-parameter generalised beta distribution is easy adjustment of the probability function shape. In Figure 8, the results of the computer simulation of fibre spacing based on three-parameter generalised beta distribution are presented. The graphical representation of fibre spacing shown in Figure 8 is difficult to analyse, especially in terms of fibre concentration fields and lack of fibre fields. Due to the fact that uniformity of fibre spacing is essential to the safety of SFRC structure, one should look for fibre concentration fields and lack of fibre fields. A special spot map was prepared (see Figure 9) to clearly expose fibre concentration fields. 

There was noticeable conformity of displaying steel fibre spacing obtained by the radar-based technique (Figure 5) and by the computer simulation (Figure 9). In the red fields, steel fibre was highly concentrated. The similarity of the two pictures led to the conclusion that the testing equipment based on radar waves used for fibre detection is able to recognize fibre concentration fields but not a single fibre.

## 5. Conclusions

The results obtained during the research programme allowed us to form the following conclusions:The method based on the electromagnetic induction technique can be applied to estimate the approximate volume of steel fibre in a hardened SFRC element and the uniformity of fibre spacing. However, the method requires calibration to obtain good quality of results in deeper layers due to the “shadow” cast by the fibre present in the top layers. The method can be applied to detect steel fibre up to the 120 mm thickness of the tested element.The method based on the radar technique is suitable for instant detection of the areas with a clearly spaced fibre volume (too low or too high local fibre concentration). Theoretically, the method can be applied to detect steel fibre presence up to the 200 mm thickness of the tested element, yet only fibre present in upper layers is correctly detected. The testing equipment based on the radar technique used for fibre detection is able to recognize fibre concentration fields but not a single fibre.Both methods together can detect fibre concentration in SFRC volume but cannot detect a single fibre.

## Figures and Tables

**Figure 1 materials-12-03507-f001:**
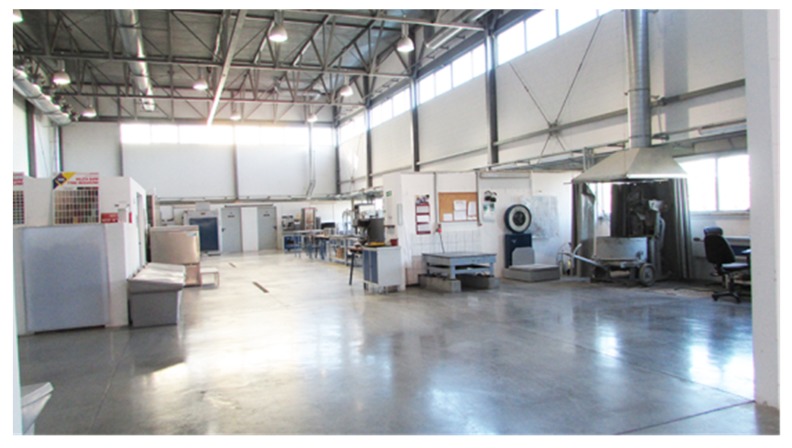
The tested industrial floor.

**Figure 2 materials-12-03507-f002:**
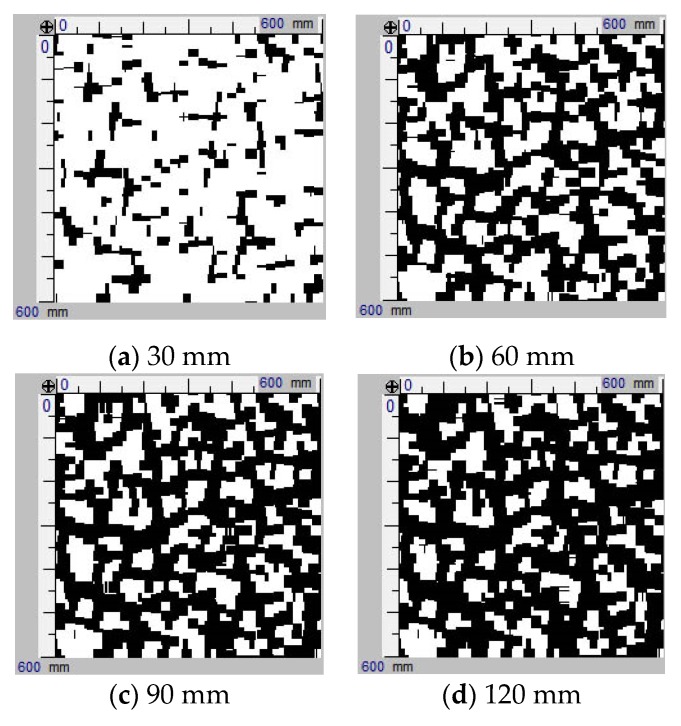
Exemplary images of a scanned area (600 mm · 600 mm) at various depths using the electromagnetic induction technique.

**Figure 3 materials-12-03507-f003:**
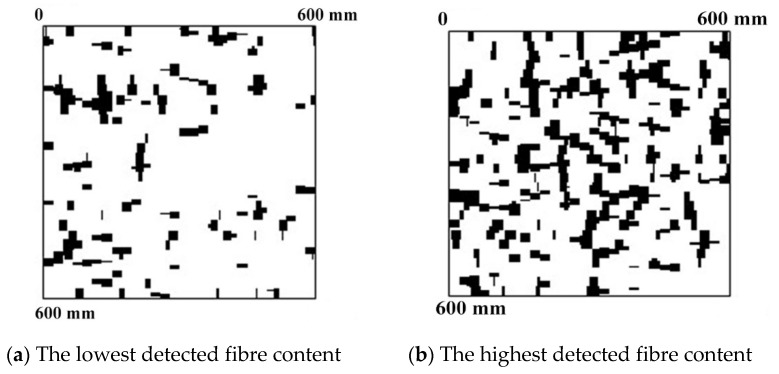
Images of detected fibre (using the electromagnetic induction technique) in different tested areas at the same depth of 30 mm.

**Figure 4 materials-12-03507-f004:**
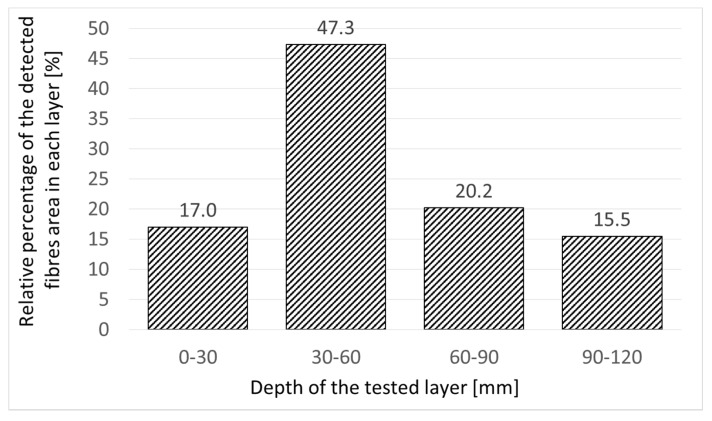
Relative percentage of the detected fibre at each scanned depth.

**Figure 5 materials-12-03507-f005:**
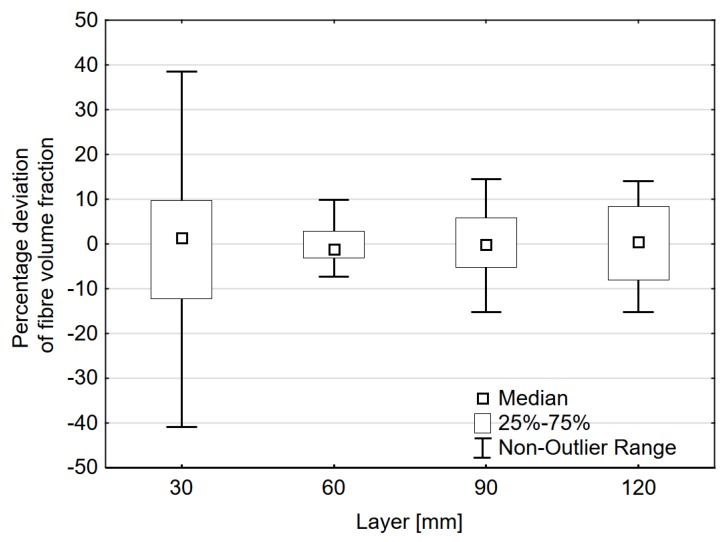
Percentage deviation of fibre volume.

**Figure 6 materials-12-03507-f006:**
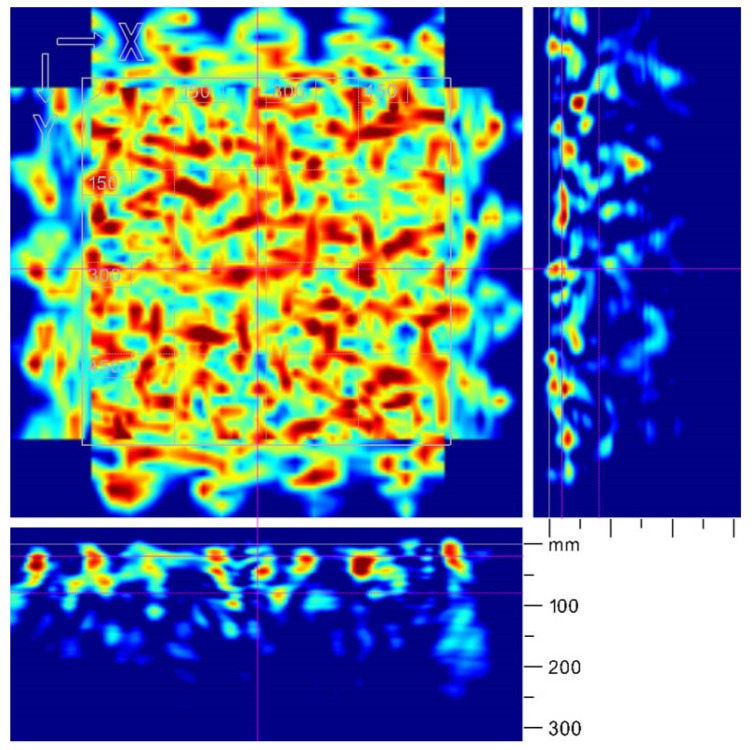
Two-dimensional visualisation of data obtained by the radar-based technique.

**Figure 7 materials-12-03507-f007:**
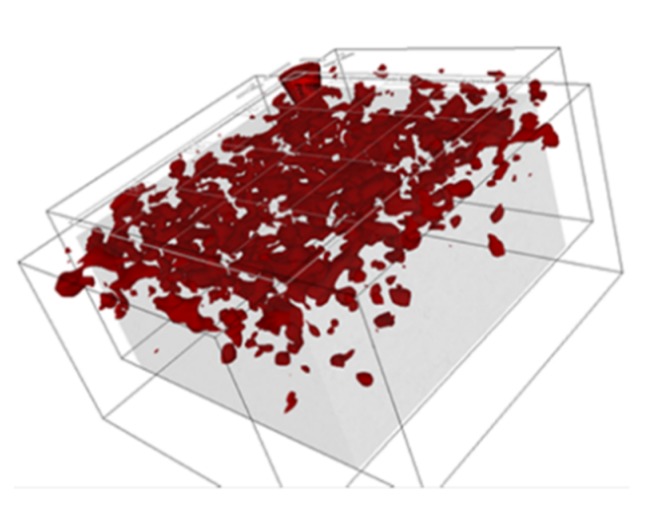
Three-dimensional visualisation of data obtained by the radar-based technique.

**Figure 8 materials-12-03507-f008:**
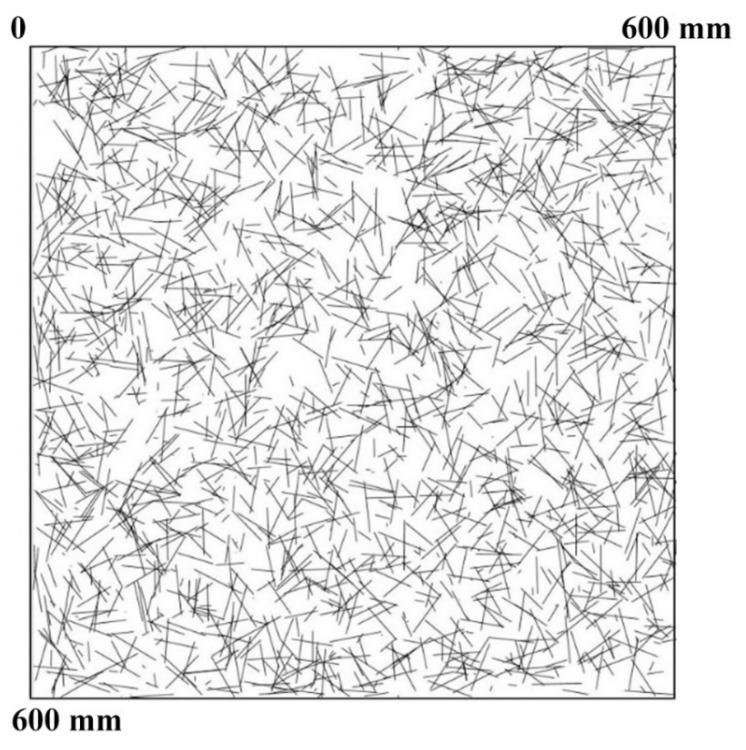
A computer simulation of fibre spacing based on the three-parameter generalised beta distribution at the depth of 30 mm.

**Figure 9 materials-12-03507-f009:**
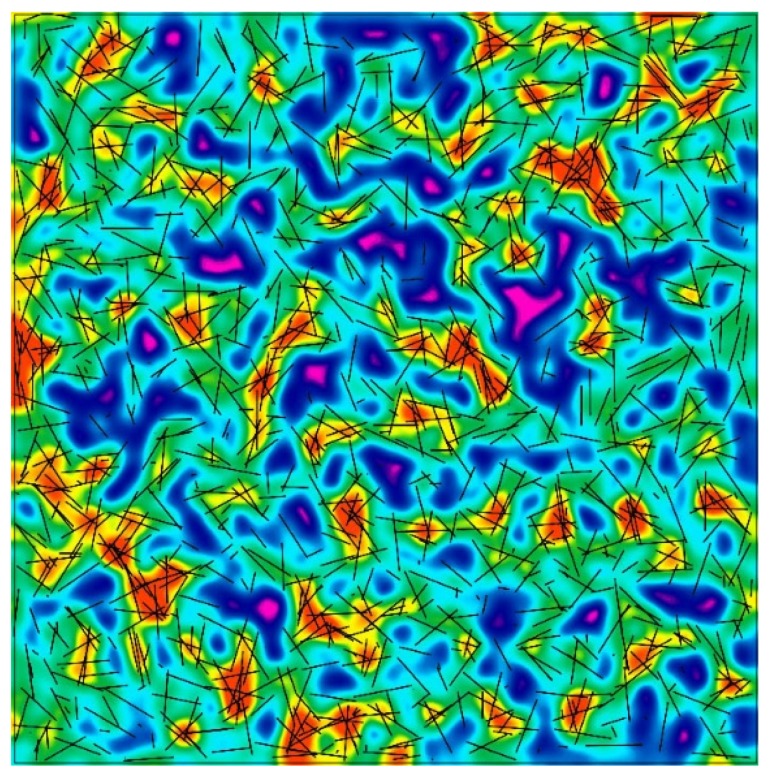
A computer simulation of fibre spacing exposing fibre concentration fields (warm colours) and low intensity of fibre appearance (cold colours).

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
