# Peer review of "A Combined Electromagnetic Induction and Radar-Based Test for Quality Control of Steel Fibre Reinforced Concrete"

_materials, 2019, doi:10.3390/ma12213507_

Round 1

Reviewer 1 Report

The authors compare two methods for detecting the fibre content and spacing in the concrete industrial floors. The article is fairly interesting for the readers of the journal. Just a remark: the authors should best explain if the so different fibre content at the different depth depend on the testing system or it is  actual.

Author Response

Dear Editor,

I am sending to you the amended version of my paper entitled: “A combined electromagnetic induction and radar-based test for quality control of steel fibre reinforced concrete” materials-619199 (co-author: Janusz Kobaka and Jacek Katzer). Amendments (marked in red and highlighted in yellow) were introduced according to reviewers’ comments.  I would like to thank the reviewers for their critique and remarks which helped to improve the paper. Detailed answers to reviewers’ comments are as follows:

Reviewer #1:

“Just a remark: the authors should best explain if the so different fibre content at the different depth depend on the testing system or it is  actual.”

Some comments were added in section 3 (lines 122 – 126) to make this matter clear for a reader.

Reviewer 2 Report

The paper also looks promising, and is of an interesting read. But however the Radar technique proves to have a serious limitation. And hence effectively this paper is not a comparison between the two types of NTD techniques. Secondly if the results of the electromagnetic induction were proved by another retrospective method, then this publication would have achieved its intention. Sadly without the intent of this paper is not served. Hence I would suggest this paper to be revised significantly for further consideration. 

Language requires minor polishing otherwise which it is of sufficient standard.

Author Response

Dear Editor,

I am sending to you the amended version of my paper entitled: “A combined electromagnetic induction and radar-based test for quality control of steel fibre reinforced concrete” materials-619199 (co-author: Janusz Kobaka and Jacek Katzer). Amendments (marked in red and highlighted in yellow) were introduced according to reviewers’ comments.  I would like to thank the reviewers for their critique and remarks which helped to improve the paper. Detailed answers to reviewers’ comments are as follows:

Reviewer #2:

“The paper also looks promising, and is of an interesting read. But however the Radar technique proves to have a serious limitation.”

Authors fully agree with the Reviewer that the radar technique has significant limitations. Therefore it can’t be used effectively (in case of SFRC) alone. The main idea of the conducted analysis was to combine two existing and commonly used NDT methods (both with their own limitations) to achieve satisfactory assessment of fibre spacing. 

“And hence effectively this paper is not a comparison between the two types of NTD techniques.”

Yes – this paper IS NOT a comparison study of the two methods. It is a proposition of using them both at the same time to achieve much more precise fibre spacing assessment than in case of utilizing only one of them. See lines 55-56 of the text where authors clearly state: “Authors decided to conduct a research programme focused on using both techniques simultaneously to assess fibre spacing in hardened SFRC.”. One has to also remember that: “When you make a comparison, you say that one thing is like another in some way” (countable noun), but also ”When you make a comparison, you consider two or more things and discover the differences between them” (variable noun) – see Collins dictionary.

 “Secondly if the results of the electromagnetic induction were proved by another retrospective method, then this publication would have achieved its intention. Sadly without the intent of this paper is not served. Hence I would suggest this paper to be revised significantly for further consideration.”

The achieved results were compared with cored samples of the SFRC (see comments in line 123 – 125). For obvious reasons the number and location of cored samples was significantly limited in comparison to number and location of NDT tests. Cored samples enabled the general proof of concept of the proposed “combined” methodology but need for further research in this area was recognized by authors.  

“Language requires minor polishing otherwise which it is of sufficient standard.”

The paper underwent language polishing. Some minor linguistic changes are marked in the text.

I look forward to hearing from you.

Best regards,

Tomasz Ponikiewski

Round 2

Reviewer 2 Report

Thank you for addressing the comments.